# Controversial Roles of Autophagy in Adenomyosis and Its Implications for Fertility Outcomes—A Systematic Review

**DOI:** 10.3390/jcm13247501

**Published:** 2024-12-10

**Authors:** Julie Vervier, Marlyne Squatrito, Michelle Nisolle, Laurie Henry, Carine Munaut

**Affiliations:** 1Laboratory of Tumor and Development Biology, Giga-Cancer, University of Liège, 4000 Liege, Belgium; 2Obstetrics and Gynecology Department, University of Liège-Citadelle Site, 4000 Liege, Belgium; 3Center for Reproductive Medicine, University of Liège-Citadelle Site, 4000 Liege, Belgium

**Keywords:** macroautophagy, adenomyosis, endometriosis interna

## Abstract

**Background/Objectives:** Adenomyosis is a benign condition where ectopic endometrial glandular tissue is found within the uterine myometrium. Its impact on women’s reproductive outcomes is substantial, primarily due to defective decidualization, impaired endometrial receptivity, and implantation failure. The exact pathogenesis of the disease remains unclear, and the role of autophagy in adenomyosis and its associated infertility is not well understood. The aim of this systematic review was to conduct an exhaustive search of the literature to clarify the role of autophagy in the pathogenesis of adenomyosis. **Methods:** A systematic search was conducted in Medline, Embase, and Scopus databases up to the date of 20 August 2024. We included all English-written publications assessing the role of autophagy in the pathogenesis of adenomyosis. **Results:** Seventeen eligible articles were identified, including reviews and experimental studies involving human samples and murine models. The results showed that the role of autophagy in adenomyosis is controversial, with studies showing both increased and decreased levels of autophagy in adenomyosis. **Conclusions:** Autophagy plays a dual role in cell survival and death. Increased autophagy might support the survival and proliferation of ectopic endometrial cells, while decreased autophagy could prevent cell death, leading to abnormal growth. Oxidative stress may trigger pro-survival autophagy, mitigating apoptosis and promoting cellular homeostasis. Hormonal imbalances disrupt normal autophagic activity, potentially impairing endometrial receptivity and decidualization and contributing to infertility. The balance of autophagy is crucial in adenomyosis, with its dual role contributing to the complexity of the disease. Limitations: A few studies have been conducted with heterogeneous populations, limiting comparative analyses.

## 1. Introduction

Adenomyosis is a benign condition characterized by the presence of ectopic endometrial glandular tissue within the uterine myometrium and surrounded by a smooth muscle hyperplasia reaction [1]. The disease affects 20% of women of reproductive age [2] and often coexists with endometriosis (20–80%) and/or uterine fibroids (15–57%) [1,2,3]. While some patients can be asymptomatic, most of the women suffering from adenomyosis will experience abnormal uterine bleeding, pelvic pain, or infertility [4]. Concerning infertility, adenomyosis is strongly associated with reduced pregnancy rate and live birth rate after ART with a reduction of 31–43% in the likelihood of conception. Increased miscarriage rates are also observed for those patients suffering from adenomyosis [5]. The impact of adenomyosis on fertility is mostly due to defective decidualization, impaired endometrial receptivity, and implantation failure. The precise pathogenesis is still poorly understood, although the involvement of uterine hyperperistalsis, abnormal endometrial steroid metabolism, increased intrauterine oxidative stress, impairment of implantation factors, reduced gut microbiota diversity, and occurrence of chronic endometritis have been described [6,7].

The pathogenesis of adenomyosis itself is also poorly understood, with probable involvement of sex steroid hormone receptors, inflammatory molecules, extracellular matrix enzymes, growth factors, and proangiogenic factors. Recently, the involvement of autophagy in the disease progression and its role in adenomyosis-associated infertility have been discussed [8].

Autophagy is an intracellular mechanism involved in recycling aging organelles and non-functional proteins. Initially described as a phenomenon activated under stress or fasting conditions (non-selective autophagy), it is now accepted that a basal level of autophagy is essential for several cellular functions to maintain cellular homeostasis (selective autophagy) [9,10]. The endometrium undergoes variations in autophagy during the menstrual cycle under the influence of hormonal changes [11]. In addition to the menstrual cycle, autophagy has also been described in endometrial receptivity and implantation. Autophagy is essential for cell survival and tissue growth, and alterations in the autophagic pathway have been described in endometrial pathologies such as endometrial hyperplasia and carcinoma, infertility, and endometriosis [10].

In endometrium-related pathologies, autophagy plays a dual role, yet our understanding of its multifaceted role still needs to be improved. Specifically, there is a scarcity of studies investigating the role of autophagy in adenomyosis [12]. There is an ongoing debate about whether autophagy is upregulated or downregulated in adenomyosis. Further exploration into the role of autophagy in the disease’s progression and its role in the infertility mechanisms is essential.

The aim of this systematic review was to conduct an exhaustive search of the literature in order to clarify the role of autophagy in the pathogenesis of adenomyosis.

## 2. Materials and Methods

A search to identify pertinent studies was performed through the Medline (PubMed interface), Embase, and Scopus databases from 1 January 2010 up to 20 August 2024. The limitation to the last 14 years is due to the fact that the issue of autophagy modulation in adenomyosis appeared in the scientific literature only since 2010. Our search did not yield any studies published on the issue before 2010. The search strategy was based on the keywords ‘adenomyosis’ and ‘autophagy’. All studies published in English were included. They were uploaded into the Covidence software www.covidence.org (Veritas Health Information, Melbourne, Australia), and duplicates were removed. Two independent reviewers screened for the assessment of the article first by titles and abstracts. The retrieved studies were assessed in detail against the inclusion criteria, and reasons for exclusion are listed in the flow diagram (Figure 1). The bibliographies of the relevant publications identified were examined for additional articles, which could satisfy inclusion criteria. After the systematic search, articles recognized as relevant were analyzed in more detail by the main author for data collection. Data extraction was synthesized in a chart by sorting the experimental results by activation, inhibition, or no changes in autophagy. For each study, the following information was extracted: experimental methods, types of samples, origin of the samples, mean age, menstrual phase, autophagy marker studied, and outcome. When not available, the missing data were noted as not indicated (NI). The review was written according to the PRISMA 2020 statement.

## 3. Results

### 3.1. Publication Screening

Figure 1 presents the flow diagram of the systematic search. In total, 17 eligible articles were identified, including two review articles [12,13], namely, 1 experimental study using a murine model [14], 2 experimental studies using both human samples and murine models [15,16], and 12 experimental studies using human samples [17,18,19,20,21,22,23,24,25,26,27,28]. The characteristics and outcomes of the experimental articles, sorted by whether results advocate for an activation, an inhibition, or no changes in autophagy, are summarized in Table 1. This systematic review of literature highlights the controversial role of autophagy in the pathogenesis of adenomyosis. While some studies advocate for increased autophagy levels, others suggest a decrease in autophagy among adenomyosis patients. Notably, only one study did not identify a modulation of autophagy.

Several hypotheses can explain how increased or decreased autophagy could lead to the development of adenomyosis and are described below. In some settings, increased autophagy may contribute to the survival and proliferation of ectopic endometrial cells, whereas in other settings, decreased autophagy could result in resistance to cellular death, leading to abnormal cell growth and survival. In the following, we explore the characteristics of the studies advocating either for an activation or for an inhibition of autophagy in adenomyosis.

### 3.2. Activation of Autophagy

Four studies have demonstrated the activation of autophagy in adenomyosis. Some authors observed an increased expression of LC3B and Beclin1 in the eutopic endometrium of patients with adenomyosis, mostly during the proliferative phase [23,25]. One study also noted this increase during the secretory phase [26]. The proposed mechanisms behind this activation of autophagy include decreased Indian Hedgehog signaling [23], activation of PARP1/SIRT1 signaling via iron overload [25], and reduced expression of GRIM-19, leading to increased activation of the AMPK/ULK1 pathway [26].

One study specifically investigated the role of mitophagy in adenomyosis, demonstrating an increased expression of PINK1 in the endometrium of patients with adenomyosis and in the uterus of a murine model of the disease [16]. However, it is important to point out that this study involved only three patients per group, which presents a significant bias.

### 3.3. Inhibition of Autophagy

Seven studies advocate for an inhibition of autophagy in adenomyosis. In murine models, levels of LC3B and Beclin1 were found diminished in mice with adenomyosis [14,15]. This inhibition of autophagy levels was reversed with GnRHa treatment [14].

In human samples, a decrease in Beclin1 expression was observed in the eutopic endometrium of women with adenomyosis [21]. Additionally, LC3A expression was found to be diminished during the secretory phase and menstruation in the glandular compartment of the eutopic endometrium of women with adenomyosis, with no significant changes noted during the proliferative phase. In healthy endometrium, a rhythmic expression of autophagy is conserved with elevation of autophagy levels during the secretory and menstrual phases and a decrease in these levels during the proliferative phase. This loss of rhythmic LC3A expression in adenomyosis patients was linked to a decrease in TSC2 expression, which in turn would inhibit autophagy through the activation of the mTOR signaling pathway [24].

In infertile patients with adenomyosis, a reduction in the expression of LC3B and Beclin1 in the eutopic endometrium during the secretory phase has also been observed when compared to patients undergoing IVF who obtained successful pregnancies [15]. Another study corroborates these findings and describes that the expression levels of LC3A/B, Beclin1, ATG5, and ATG12 in eutopic endometrium were lower compared to women with tubal infertility [28]. The mechanisms thought to be responsible for this are impaired decidualization due to decreased autophagy levels, which could imply a decrease in KLF4 levels [15] or the mTOR pathway [28].

When comparing autophagy levels between the ectopic and eutopic endometrium of affected patients, it appears that autophagy is lower in the ectopic tissues [17,18]. While some authors showed that LC3A expression in adenomyosis was also lower in glandular ectopic tissue compared to glandular eutopic tissue [24]. Other publications showed that the expression of LC3B in the stromal compartment of the endometrium was weaker in ectopic tissue compared to eutopic tissue but showed no significant differences in glandular compartments [19,27]. Moreover, when tested, this decreased expression of LC3B in the ectopic endometrium is not significant when compared to healthy endometrium [27].

Finally, treatment with traditional Chinese medicine resulted in a greater number of autophagosomes compared to untreated adenomyotic tissue. Advocating for an inhibition of autophagy in adenomyosis [22].

In addition to the experimental studies exploring the role of autophagy in adenomyosis, two reviews were published on the subject, both advocating for an inhibition of autophagy in adenomyosis [12,13].

## 4. Discussion

Autophagy plays a multifaceted role in the uterus, particularly in the context of adenomyosis, where its regulation is crucial. From its modulation during the menstrual cycle to its dual effects in adenomyosis—either as a protective mechanism or a contributing factor to pathology—autophagy is deeply interrelated with oxidative stress, sex steroid hormone imbalances, and the essential processes of endometrial receptivity and implantation. Understanding these dynamics is key to addressing the fertility issues associated with adenomyosis.

### 4.1. Molecular Mechanism of Autophagy

Autophagy is a crucial cellular process for maintaining homeostasis, which involves well-coordinated stages with various specific proteins and protein complexes [29]. During macroautophagy, a double-stranded membrane (phagophore) progressively encompasses part of the cytoplasm containing the degradation products. This membrane progressively expands until it closes and forms the autophagosome. The latter will then fuse with the lysosome and become an autolysosome in which the lysosomal enzymes will degrade the contents [9,30]. These different steps, as illustrated in Figure 2, are divided as follows: initiation, nucleation, elongation, closure/maturation, fusion, and degradation [12].

Thus, the process begins with the initiation stage, where the ULK complex, composed of the protein kinase ULK1/2, ATG13, ATG101, and FIP200, is formed [31,32]. This complex is essential for the recruitment and activation of the class III phosphatidylinositol-3-kinase complex (PI3KC3), which includes Beclin-1, VPS34, VPS15, and ATG14. The activation of this complex leads to the local production of phosphatidylinositol 3-phosphate (PI3P), which plays a crucial role in the formation of the phagophore membrane by modifying the ER membrane and recruiting PI3P-binding proteins [31,33]. These proteins are involved in ATG9 recycling and membrane maturation, processes that occur during the elongation phase [34].

During elongation, two ubiquitin-like conjugation systems are necessary for the extension of the phagophore membrane. The ATG12-ATG5-ATG16L complex primarily localizes on the outer side of this membrane, where it facilitates the conjugation of the protein LC3 to the phosphatidylethanolamine (PE), forming LC3-II [34,35]. LC3-II integrates into both the inner and outer membranes of the autophagosome, serving as a commonly used marker for identifying these double-membrane organelles [36].

The maturation/closure stage involves the complete formation of the autophagosome, with LC3-II remaining associated with the membranes until the autophagosome fuses with a lysosome. Lysosomal enzymes then degrade the contents of the autophagosome, allowing the cell to recycle the degraded components [30].

This process is tightly regulated by mTORC1, which, when activated by growth signals and adequate nutrient levels, inhibits autophagy [37,38]. Conversely, when mTORC1 is inhibited, such as during amino acid deficiency or through AMPK activation, autophagy is initiated, enabling the cell to recycle its components to survive under stress conditions [39,40].

The autophagy pathway described above is known as the “canonical” pathway, which operates at a baseline level in all eukaryotic cells to maintain homeostasis. However, there are also non-canonical pathways where some ATG proteins, such as Beclin-1, are not required for autophagosome formation. For instance, autophagy has been shown to be regulated by Beclin-1 through the MAPK/ERK and MAPK/JNK pathways through the BCL-2 anti-apoptotic protein [41,42,43].

In summary, autophagy is a complex and extremely regulated process, essential for cellular balance and stress response, with numerous protein complexes playing central roles. These proteins are critical not only in autophagy but also in other cellular functions.

### 4.2. Modulation of Autophagy During the Menstrual Cycle

#### 4.2.1. Autophagy in Healthy Endometrium

The endometrium is a tissue that undergoes a variation in autophagy during the menstrual cycle under the influence of hormonal variations [10]. As illustrated in Figure 3, autophagic activity within the endometrium is closely related to cell proliferation and apoptosis, thus playing a crucial role in the dynamic refreshment of the endometrium during each menstrual cycle [12].

During the proliferative phase, under the influence of estrogens, autophagy is inhibited, allowing cell proliferation. Indeed, Choi et al. observed that endometrial stromal cells (ESCs) cultured with estrogen alone to mimic the proliferative phase showed a significant decrease in LC3-II expression levels compared to ESCs cultured with both estrogen and progesterone [44].

Autophagic activity gradually increases in the endometrium during the secretory phase under the influence of progesterone, peaking in the late secretory phase. It has been demonstrated that progesterone significantly increased ATG5 and LC3-II expression in estrogen-treated ESCs in vitro while simultaneously leading to a significant decrease in AKT phosphorylation, which negatively regulated autophagy induction by activating mTOR (Figure 1) [45]. This rise in autophagosomes leads to apoptosis through alterations in the BAX:BCL2 ratio and subsequent caspase 3 activation, ultimately inducing desquamation of the functional layer of the endometrium, resulting in menstruation [11].

#### 4.2.2. Influence of Sex Steroid Hormone Imbalance

Steroid hormones are known to be involved in the pathogenesis of adenomyosis. Local hyperestrogenic levels within the ectopic endometrial lesions are reported [8], which induces a downregulation of progesterone receptors responsible for progesterone resistance [46].

As autophagy within the endometrium is strongly correlated to hormonal variations, the abnormal hyperestrogenic status could lead to altered physiological variation in autophagy in adenomyosis (Figure 4). Indeed, in vitro experiments from adenomyosis tissues observed a loss of the rhythmic expression of autophagy levels, leading to decreased autophagy [27]. Furthermore, in vivo, gonadotrophin-releasing hormone agonist (GnRHa) treatment restores autophagy levels, probably due to the drug’s anti-estrogenic effect [14].

Notably, the variation in autophagy reported in the papers selected by this systematic review seems to be linked to the menstrual phase. Most papers showing inhibition of autophagy took samples from the secretory phase, while those showing activation of autophagy had samples from the proliferative phase [23,25]. Two papers that took samples from both phases showed conflicting results, with one advocating for increased autophagy [26] and the other for the opposite [27]. These results suggest the hypothesis that the autophagic imbalance may fluctuate throughout the menstrual cycle, implying that the primary factor in pathogenesis might not be the increase or decrease in the phenomenon, but rather the impairment of autophagy regardless of its trend. Notably, the abstracts published by Popryadukhin [17,18] highlight a decrease in autophagy in the ectopic endometrium compared to the eutopic endometrium of patients with adenomyosis during the proliferative phase. However, these results were not compared to healthy controls, making the interpretation of modulation of autophagy depending on the phase of the menstrual cycle biased.

### 4.3. Is Autophagy Up- or Downregulated in Adenomyosis?

#### Autophagy’s Double-Faced Role

Selective autophagy is involved in processes such as the regulation of cell death, cell proliferation, inflammation, and innate or adaptive immune functions. The absence or alteration of this autophagy process has been linked to the development of numerous pathologies, including endometrial hyperplasia, carcinoma, infertility, and endometriosis. Research on autophagy in pathological processes is complicated by the fact that autophagy is a “double-edged sword” that can be either protective or harmful depending on the biological context [12]. Maintaining cellular homeostasis necessitates a basal level of autophagy. In stressful conditions, a deficiency in autophagy can result in cell death, as autophagy plays a protective role under such circumstances. Conversely, hyperactivated autophagy can also lead to cell death [47]. An imbalance of autophagy, whether favoring underexpression or overexpression, leads to cellular dysfunction and contributes to pathological development. This requirement for a precise balance of autophagy may elucidate the contradictory outcomes reported in studies investigating the role of autophagy in adenomyosis pathogenesis.

On one hand, reduced autophagy levels promote cell survival and proliferation. Beclin1 is a major autophagy-related protein acting at a cross point of autophagy and apoptosis by binding with the anti-autophagy and anti-apoptotic protein Bcl-2. Within the endometrium, in vivo and in vitro experiments have demonstrated the inverse correlation between Beclin1 expression and cell proliferation. Decreased Beclin1 expression is associated with increased cellular proliferation, migration, and EMT through the activation of the PI3K/AKT/mTOR axis, and an overexpression of Beclin1 expression reduces cellular proliferation and enhances apoptosis [24,48].

On the other hand, regenerating nutrients by recycling metabolic material could facilitate cell survival, proliferation, and invasion. Since adenomyosis shares certain similarities with tumorous lesions—such as invasion, adhesion, relapse, and angiogenesis—autophagy may act similarly to its role in cancer development. Research supports that the downregulation of the tumor suppressor gene GRIM-19 in adenomyosis increases autophagy levels via the AMPK/ULK1 signaling pathway, allowing adenomyosis nidus to generate the energy and nutrients necessary for its survival and growth [26].

### 4.4. The Role of Oxidative Stress

Under such stressful conditions, autophagy can be activated, potentially triggering a pro-survival mechanism that mitigates oxidative stress-induced cell apoptosis and helps restore cellular homeostasis [49]. In adenomyosis lesions, iron overload resulting from repeated hemorrhages is common and contributes to a disrupted local environment that induces oxidative stress through the generation of ROS [50]. Under such stressful conditions, autophagy can be activated, potentially triggering a pro-survival mechanism that mitigates oxidative stress-induced cell apoptosis and helps restore cellular homeostasis [51]. This cytoprotective autophagy activity could be activated by inhibiting the Indian hedgehog (Ihh) pathway or PARP1/SIRT1 signaling pathways [23,25].

A key aspect of this response involves mitophagy, a selective autophagy process that specifically targets damaged mitochondria [52]. The link between oxidative stress and mitophagy is crucial, as mitochondria are both a source and target of ROS. When oxidative stress damages mitochondria, they can become dysfunctional, producing even more ROS and exacerbating cellular damage. To counteract this, cells initiate mitophagy to remove these damaged mitochondria, thereby reducing the overall level of oxidative stress and maintaining cellular function [53]. However, in the context of adenomyosis, this process can become dysregulated. Indeed, when this mitophagy becomes excessively active due to persistent oxidative stress, as seen in adenomyosis, it leads to the elimination of damaged mitochondria rather than their repair. This overactive process results in a decline in mitochondrial numbers, disrupting cellular function and promoting cellular invasiveness [16].

Evidence showed that reducing PINK1 levels, a mitophagy regulator, significantly diminishes the invasion potential of endometrial stromal cells (ESCs) in adenomyosis [16]. The pathological progression of endometrial invasion into the myometrium in adenomyosis correlates with the accumulation of reactive oxygen species (ROSs) [16,25]. While this accumulation may influence rather the cell survival or the abnormal invasion capability observed in adenomyosis, further investigation is needed to elucidate the specific mechanisms involved.

In light of these findings, autophagy activation could be viewed as an adaptive response in promoting cell survival or cell invasion under stressful conditions.

### 4.5. Impact on Endometrial Receptivity and Implantation

Autophagy plays a crucial role in endometrial receptivity and embryo implantation, both of which are essential processes in mammalian reproduction. The study by Su et al. demonstrated that the expression of autophagy markers, such as ATG5 and LC3, decreases after embryo attachment, particularly at the implantation site. Inhibition of autophagy in mice resulted in fewer implantation sites and disrupted uterine decidualization, suggesting that autophagy is vital for embryo implantation and is likely involved in endometrial decidualization during early pregnancy [54].

Optimal autophagy activity in the endometrium is essential for mitigating cellular stress and controlling inflammation, which are essential for successful embryo implantation and pregnancy. Disruptions in autophagy can impair endometrial receptivity, leading to fertility issues. Understanding this interaction could lead to new therapeutic approaches for reproductive disorders by enhancing endometrial support for embryos [55].

In the context of adenomyosis, a condition often associated with infertility, implantation failure and early miscarriages are common, largely due to defective decidualization of endometrial stromal cells [10]. Women experiencing repeated implantation failure often show decreased autophagy levels in the endometrium, linked to a reduction in decidualization capacity [56].

Studies on adenomyosis patients and mouse models have revealed lower autophagy levels in the eutopic endometrium during the decidual phase and the implantation window compared to controls [15,28] This decrease in autophagy may be due to the abnormal expression of KLF4 in endometrial stromal cells, a factor known to regulate ATG5 expression and maintain normal autophagy and decidualization processes [15].

The importance of autophagy in decidualization is further supported by research on the autophagy gene ATG16L1. While hypomorphic ATG16L1 mice were fertile and showed normal decidualization, conditional knockout of ATG16L1 in the reproductive tract led to reduced fertility due to a lower implantation rate. In the absence of ATG16L1, endometrial stromal cells failed to properly decidualize, resulting in fewer blastocysts implanting. Similarly, knocking down ATG16L1 in human ESCs adversely affected decidualization. These findings suggest that ATG16L1 is essential for decidualization, implantation, and overall fertility in mice and may also play a crucial role in implantation success in women [57].

Moreover, systemic knockout studies have shown that various ATG-related genes, such as Beclin-1, RB1CC1 (FIP200), and AMBRA1 (a regulator of Beclin-1), are essential for embryonic development, and their inefficiency leads to embryonic lethality and severe developmental defects [58]. Despite this, their specific roles in embryo implantation are not well understood. Recent research has highlighted the critical involvement of autophagy in endometrial cells for successful embryo implantation and decidualization in women [57]. For instance, Oestreich et al. used a conditional knockout mouse model for RB1CC1 to show that the autophagy protein FIP200 is vital for the transformation of endometrial stromal cells into decidualized cells, emphasizing the importance of autophagy in these processes [59].

These insights underscore the pivotal role of autophagy, particularly via regulators like KLF4 and genes such as ATG16L1, in successful embryo implantation. They also highlight the potential for targeting autophagic pathways to improve fertility treatments, especially in conditions like adenomyosis where these processes are dysregulated.

### 4.6. Potential Therapeutic Implications of Targeting Autophagy in Adenomyosis

Understanding the multifaceted role of autophagy in adenomyosis opens avenues for potential therapeutic interventions, particularly in addressing fertility issues associated with the condition. Modulating autophagy pathways could offer novel strategies for both the treatment and diagnosis of adenomyosis-related infertility. Autophagy plays a critical role in maintaining cellular homeostasis and has been recognized as a key player in the pathophysiological process related to the endometrium-related diseases such as endometriosis, endometrial carcinoma, and infertility [10,60]. Therapeutically, targeting the autophagy pathway may offer a promising avenue for intervention. Pharmacological agents such as rapamycin, an mTOR inhibitor that enhances autophagy, or chloroquine/hydroxychloroquine, which inhibit autophagy, have been explored as potential treatments in similar gynecological diseases [54,61,62].

Indeed, preclinical studies have shown that activation of autophagy by rapamycin can reduce the size of endometriotic lesions, suggesting a similar therapeutic potential in adenomyosis [63].

Moreover, diagnostic strategies could leverage biomarkers of autophagic activity, such as LC3, Beclin-1, and p62, to better stratify patients and personalize therapeutic approaches [64]. Assessing these biomarkers in endometrial tissues could provide insights into the autophagic status and guide clinical decisions.

These perspectives open new avenues for integrating autophagy-targeting therapies into the clinical management of adenomyosis-related infertility. However, further preclinical and clinical studies are necessary to validate these approaches and determine their safety and efficacy.

### 4.7. Future Directions

The contradictory role of autophagy in adenomyosis underscores the need for further research to clarify its mechanisms and therapeutic potential. Future studies should monitor autophagy levels throughout the menstrual cycle in both eutopic and ectopic endometrial tissues to determine whether autophagic imbalance is consistent or varies with hormonal fluctuations. Elucidating the molecular mechanisms underlying autophagy dysregulation using advanced molecular techniques—such as transcriptomics, proteomics, and metabolomics—is crucial. Understanding the specific signaling pathways and genes involved may reveal novel therapeutic targets and enhance our comprehension of the disease’s etiology.

An emerging area of interest is the heterogeneity of autophagic responses among different cell types within the endometrium. Recent single-cell RNA sequencing (scRNA-seq) studies have unveiled the complex cellular composition of the endometrial tissue, including various subtypes of epithelial cells, stromal cells, immune cells, and vascular cells [65]. These cell types may exhibit distinct autophagic activities and respond differently to hormonal signals and pathological stimuli. Leveraging scRNA-seq and other single-cell technologies to dissect the heterogeneous effects of autophagy at the cellular level could uncover novel biomarkers and therapeutic targets by identifying cell populations that contribute significantly to disease progression through dysregulated autophagy.

Exploring the therapeutic potential of autophagy modulators remains crucial. Preclinical studies assessing agents like mTOR inhibitors (e.g., rapamycin) and autophagy inhibitors (e.g., chloroquine) should evaluate their efficacy in reducing lesion size, inhibiting cell proliferation, and improving fertility outcomes. Combining these agents with existing hormonal therapies might enhance treatment effectiveness and overcome progesterone resistance, potentially translating into tangible benefits for patients.

Developing autophagy-related diagnostic biomarkers, such as LC3-II, Beclin-1, and p62, offers a promising avenue for early detection and personalized treatment. Validating these biomarkers in clinical settings could facilitate disease monitoring and patient stratification, ultimately improving management strategies for adenomyosis. Additionally, understanding the interaction between autophagy and oxidative stress may provide insights into lesion progression and identify new therapeutic targets. Combining antioxidants with autophagy modulators might offer a synergistic approach to treatment, potentially alleviating symptoms and improving fertility.

Standardizing research methodologies—including unified definitions of adenomyosis types (e.g., focal vs. diffuse) and consistent techniques for measuring autophagy—is vital for advancing the field and improving the reliability of results. Ensuring that future studies specify the menstrual cycle phase during sample collection is essential, given the significant variation in autophagy throughout the cycle. By addressing these areas, future research can deepen our understanding of autophagy’s role in adenomyosis and pave the way for developing targeted therapies to improve fertility outcomes for affected women.

### 4.8. Limitations

Only a few studies have been published, and populations are heterogeneous in multiple ways. This leads to a restriction in the possibility of analyses that could be performed by pairing the results obtained between the studies. In particular, none of the published studies provided information on the type of adenomyosis (either focal or diffuse). Discriminating between types of lesions could aid in investigating the physiological processes underlying each variation. An additional limitation across the studies included in this review is the lack of hormone level measurements for participants. Given the significant role of sex steroid hormones—particularly estrogen and progesterone—in regulating autophagy, the absence of hormonal profiling may contribute to the heterogeneity of the results and impede a comprehensive understanding of autophagy modulation in adenomyosis. Future studies should include hormone detection to better elucidate the relationship between hormonal status and autophagic activity. This is particularly important since, as extensively discussed, autophagy varies considerably throughout the menstrual cycle.

## 5. Conclusions

In conclusion, the role of autophagy in adenomyosis pathogenesis presents a complex and nuanced landscape, as evidenced by this systematic review of the literature. The contradictory findings regarding autophagy modulation in adenomyosis patients underscore its double-faced nature, acting both as a protective mechanism and a promoter of cellular dysfunction. An imbalance in autophagy levels via oxidative stress may contribute to the pathological development of adenomyosis, influencing cell survival, proliferation, and invasion. Furthermore, sex steroid hormone imbalance disrupts physiological autophagic variations in the endometrium. The fluctuation of autophagic imbalance throughout the menstrual cycle suggests that the main factor of pathogenesis may rather be its dysfunction regardless of its trend. Finally, impaired autophagy in adenomyosis impacts endometrial receptivity and implantation, contributing to infertility through compromised decidualization. Further research is necessary to clarify these mechanisms and understand autophagy’s precise role in adenomyosis.

## Figures and Tables

**Figure 1 jcm-13-07501-f001:**
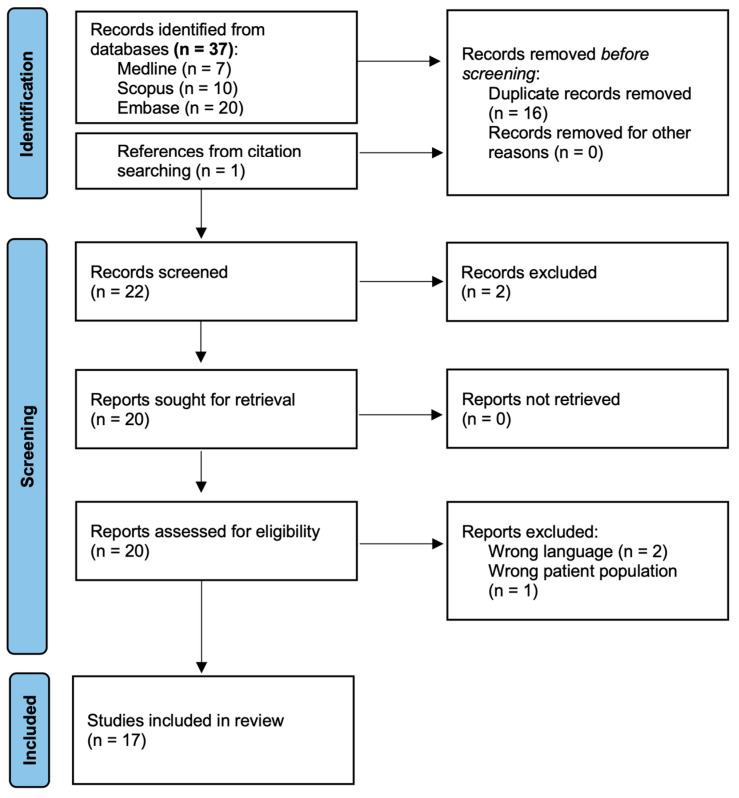
Flow diagram of the search strategy.

**Figure 2 jcm-13-07501-f002:**
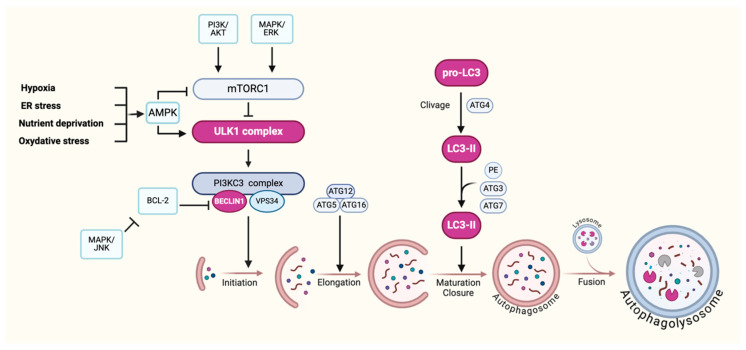
**Regulation of the autophagy pathway.** The figure illustrates the key steps of the autophagy process, highlighting molecular regulation under stress conditions, such as hypoxia, nutrient deprivation, ER stress (endoplasmic reticulum stress), and oxidative stress. AMPK (AMP-activated protein kinase) activation, through inhibition of mTORC1 (mechanistic target of rapamycin complex 1), initiates autophagy by activating the ULK1 (Unc-51 like autophagy activating kinase 1) complex, which subsequently promotes the formation of the PI3KC3 (phosphatidylinositol 3-kinase class III) complex involving Beclin1 and Vp34. The figure shows the stepwise process from the initiation of phagophore formation, elongation mediated by the ATG12-ATG5-ATG16 complex, to the maturation of the autophagosome. The processing of LC3 (microtubule-associated protein 1A/1B-light chain 3) is depicted, showing its conversion from pro-LC3 to LC3-II, essential for autophagosome formation and closure. Finally, the fusion of autophagosomes with lysosomes results in the formation of autophagolysosomes, where the degradation of cellular components occurs. Created in BioRender by Munaut, C. (2024).

**Figure 3 jcm-13-07501-f003:**
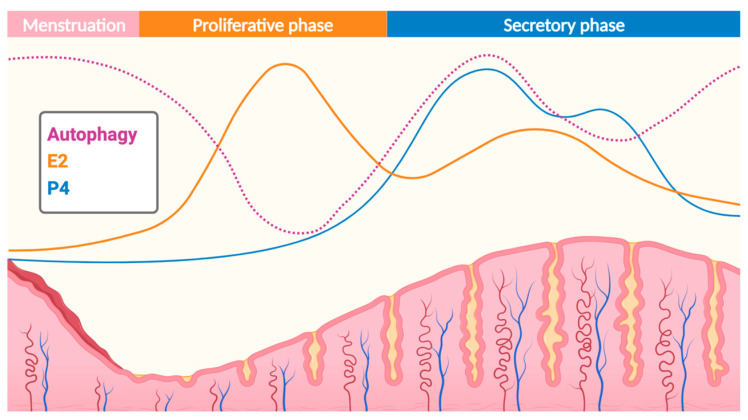
**Cyclic variation in autophagy, estradiol** (**E2**)**, and progesterone** (**P4**) **levels across the menstrual cycle.** The figure illustrates the fluctuations of autophagy (dotted pink line), estradiol (E2, orange line), and progesterone (P4, blue line) throughout the three phases of the menstrual cycle as follows: menstruation, proliferative phase, and secretory phase. During the proliferative phase, rising estradiol levels promote endometrial proliferation, correlating with a peak in autophagy activity. In the secretory phase, progesterone levels increase, preparing the endometrium for potential implantation, while autophagy activity decreases. The endometrium structure below reflects its cyclical changes, with the greatest thickness during the secretory phase. Created in BioRender byMunaut, C. (2024).

**Figure 4 jcm-13-07501-f004:**
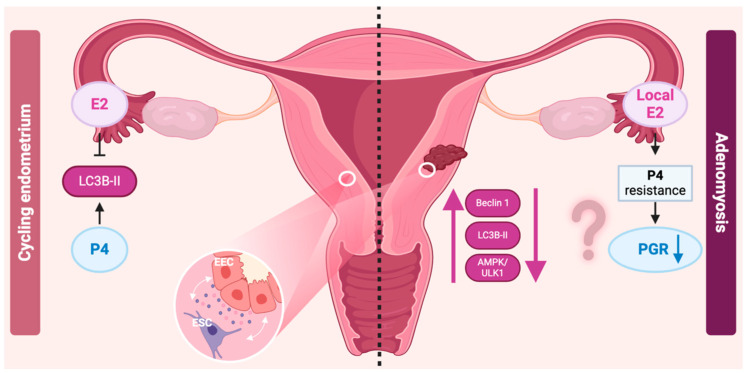
**Comparison of autophagy and hormonal regulation in cycling endometrium versus adenomyosis.** In the **cycling endometrium** (**left side**), estrogen (E2, estradiol) and progesterone (P4) regulate autophagy through the upregulation of LC3B-II (microtubule-associated protein 1A/1B-light chain 3, a key marker of autophagy). This process involves both endometrial epithelial cells (EECs) and endometrial stromal cells (ESCs), showing how the hormonal interplay influences autophagic activity in a healthy endometrial cycle. In **adenomyosis** (**right side**), local estrogen (E2) and resistance to progesterone (P4) are observed, leading to reduced expression of the progesterone receptor (PGR). This hormonal imbalance is associated with increased expression of autophagy-related markers such as Beclin 1 (a crucial autophagy initiator), LC3B-II, and AMPK/ULK1 (AMP-activated protein kinase/Unc-51 like autophagy activating kinase 1), indicating enhanced autophagic activity in adenomyotic lesions. Created in BioRender by Munaut, C. (2024).

**Table 1 jcm-13-07501-t001:** Expression and impact of autophagy in adenomyosis through synthesizing recent human and murine study outcomes.

Outcomes	M & M	Type of Samples	Sample Details	Age (Mean or Range)	Marker (s)	Reference
**ACTIVATION**						
**Elevated PINK1 mitophagy in ADM**	WB IHC TEM	hESCs from eutopic endometrium AND Entire uterus from mice	Human: 6 (3 ADM + 3 CTL) Mice: 12 (6 ADM + 6 CTL)	NI	OPTIN, PINK1, Parkin2, P62, and NDP52	Chen et al., 2023 [16]
**Indian hedgehog** (**Ihh**) **suppression activates endometrial autophagy**	WB	Eutopic endometrium	44 (18 ADM, 15 CTL, 21 ENDO) proliferative phase	25–50	LC3B	Zhou et al., 2021 [23]
**Iron overload activates autophagy**	WB	hESCs from eutopic endometrium	34 (15 ADM + 19 ENDO) proliferative phase	19–50	Beclin1 and LC3B	Zhou et al., 2022 [25]
**GRIM19 downregulation increases autophagy**	WB IHC RT-PCR	Eutopic endometrium	40 (20 ADM + 20 CTL) secretory (*n* = 19) and proliferative (*n* = 21)	39.5 (ADM) 38.4 (CTL)	Beclin1, LC3B, AMPK, and ULK1	Huang et al., 2022 [26]
**INHIBITION**						
**GnRHa reverses autophagy inhibition**	WB TEM	Eutopic endometrium from ICR mice	18 (6 ADM + placebo, 6 ADM + GnRHa, 6 CTL) implantation window		Beclin1/ß and LC3-II,	Guo et al., 2016 [14]
**Decreased KLF4 reduces autophagy and impairs decidualization**	WB IHC	hESCs and eutopic endometrium AND Entire uterus from ICR mice	Human: 24 (12 ADM + 12 CTL) secretory mice: 16	30.5 (ADM) 30.1 (CTL)	Beclin1 and LC3B	Mei et al., 2022 [15]
**Beclin1 is decreased in ADM.**	WB RTqPCR	hESCs and eutopic endometrium	62 (30 ADM + 32 CTL)	42.9 (ADM) 43.47 (CTL)	Beclin1	Ren et al., 2010 [21]
**TSC2 decrease inhibits autophagy**	WB TEM	hEECs, eutopic, and ectopic endometrium	40 (20 ADM + 20 CTL) secretory (*n* = 24) and proliferative (*n* = 16)	47.5 (ADM) 49 (CTL)	LC3B and mTOR	Gu et al., 2021 [24]
**Lower autophagy, apoptosis in ectopic ADM.**	IHC	Eutopic and ectopic endometrium	30 (15 ADM + 15 CTL) proliferative (*n* = 10), secretory (*n* = 10), and menstrual (*n* = 10)	45.2 (ADM) 43.2 (CTL)	LC3B	D’Argent et al., 2023 [27]
**Autophagy is decreased in adenomyotic stroma.**	IHC	Eutopic and ectopic endometrium	17 (8 ADM + 9 CTL)	NI	LC3B	Stratopoulou et al., 2024 [19]
**The lower decidualization is associated with its reduced autophagy induced by the mTOR pathway.**	WB	hESCs from eutopic endometrium	19 (5 ADM, 7 infertile CTL, 7 fertile CTL) secretory (*n* = 12) and decidual (*n* = 7)	NI	LC3A/B, Beclin-1, ATG5, and ATG12	Cheng et al., 2017 [28]
**Herbal treatment alters cell ultrastructure**	TEM	Primary hESCs from adenomyosis foci	6	20–50	Cytoplasmic autophagosome	Zeng et Li, 2017 [22]
**NF-kb activity may decrease autophagy in ectopic endometrium compared with eutopic endometrium.**	IHC	Eutopic and ectopic endometrium	9 proliferative	NI	LAMP1, LC3B, and NF-kp	Popryadukhin et al., 2018 [18]
**Increased beta-catenin expression could suppress autophagy in ectopic endometrium.**	IHC	Eutopic and ectopic endometrium	20 proliferative	NI	LAMP1, LC3B, and Bcl-2	Popryadukhin et al., 2020 [17]
	NO MODULATION					
**No modulation in ADM endometrium**	RTqPCR	Eutopic endometrium	32 (16 ADM + 16 CTL) proliferative	44.8 (ADM) 34.6 (CTL) ***p* < 0.001**	Beclin1, LC3B, and Bcl2	Tantanavipas et al., 2021 [20]

Abbreviations: WB: Western blot; IHC: immunohistochemistry; RTqPCR: real-time quantitative polymerase chain reaction; TEM: transmission electron microscopy; hESCs: human endometrial stromal cells; hEECs: human endometrial epithelial cells; ADM: adenomyosis; CTL: control; ENDO: endometriosis; NI: not indicated. Note: None of the studies reported hormone levels (e.g., estrogen and progesterone) of the participants.

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
