# Peer review of "Controversial Roles of Autophagy in Adenomyosis and Its Implications for Fertility Outcomes—A Systematic Review"

_jcm, 2024, doi:10.3390/jcm13247501_

Round 1
Reviewer 1 Report
Comments and Suggestions for Authors
In this review, the authors summarized the controversial roles of autophagy in Adenomyosis and Its Implications for fertility outcomes. The review is well organized and related studies are adequate. Below are a number of issues that the authors shall address or revise:
1. In Table 1, I wonder whether there were hormone detection results for each study.
2. In Table 1, the ages of volunteers were important. Do you have any age information in these studies?
3. In the discussion part, the authors can give more perspectives on cell type heterogeneous effects on autophagy. Some single-cell RNA-seq data related to endometrium may contain this information.
Author Response
In this review, the authors summarized the controversial roles of autophagy in Adenomyosis and Its Implications for fertility outcomes. The review is well organized and related studies are adequate. Below are a number of issues that the authors shall address or revise:
Thank you for your careful reading of our manuscript and for your valuable suggestions. We appreciate your positive feedback on the organization and adequacy of the related studies. We have addressed each of your comments in detail below and made corresponding revisions to the manuscript, highlighted for your convenience.
- In Table 1, I wonder whether there were hormone detection results for each study.
Thank you for bringing up this important point. We agree that hormone levels are crucial factors influencing autophagy activity in the endometrium and could significantly impact the findings related to adenomyosis.
Upon reviewing the articles included in our systematic review, we found that, unfortunately, none of the studies provided hormone detection results for their participants. We recognize that this is a limitation that could affect the interpretation of the data.
To address this, we have added a statement in the Discussion section (Section 4.8 Limitations, lines 483-489):
"A notable limitation across the studies included in this review is the lack of hormone level measurements for participants. Given the significant role of sex steroid hormones—particularly estrogen and progesterone—in regulating autophagy, the absence of hormonal profiling may contribute to the heterogeneity of the results and impede a comprehensive understanding of autophagy modulation in adenomyosis. Future studies should include hormone detection to better elucidate the relationship between hormonal status and autophagic activity."
We have now mentioned this limitation in Table 1 footnotes as followed:
"Note: None of the studies reported hormone levels (e.g., estrogen, progesterone) of the participants."
We believe that acknowledging this limitation explicitly in the manuscript adds transparency and highlights an important area for future research.
- In Table 1, the ages of volunteers were important. Do you have any age information in these studies?
We agree with your comment about the importance of including participant age information, as age can influence both autophagy processes and the prevalence of adenomyosis.
We have revisited all the studies included in our review and extracted the age information wherever it was available. We have updated Table 1 to include a new column titled "Age (mean or range)" to present this data.
- In the discussion part, the authors can give more perspectives on cell type heterogeneous effects on autophagy. Some single-cell RNA-seq data related to endometrium may contain this information.
We appreciate this insightful suggestion. The heterogeneity of cell types within the endometrium indeed plays a significant role in autophagy regulation and could contribute to the diverse effects observed in adenomyosis.
We have expanded the Discussion section to include perspectives on cell-type-specific effects on autophagy, incorporating findings from single-cell RNA sequencing (scRNA-seq) studies.
The added Text in Discussion (Section 4.7 Future Directions, lines 447-455):
“An emerging area of interest is the heterogeneity of autophagic responses among different cell types within the endometrium. Recent single-cell RNA sequencing (scRNA-seq) studies have unveiled the complex cellular composition of the endometrial tissue, including various subtypes of epithelial cells, stromal cells, immune cells, and vascular cells (Wang et al. 2020). These cell types may exhibit distinct autophagic activities and respond differently to hormonal signals and pathological stimuli. Leveraging scRNA-seq and other single-cell technologies to dissect the heterogeneous effects of autophagy at the cellular level could uncover novel biomarkers and therapeutic targets by identifying cell populations that contribute significantly to disease progression through dysregulated autophagy.”
Reference List
Wang, W., F. Vilella, P. Alama, I. Moreno, M. Mignardi, A. Isakova, W. Pan, C. Simon, and S. R. Quake. 2020. 'Single-cell transcriptomic atlas of the human endometrium during the menstrual cycle', Nat Med, 26: 1644-53.
Reviewer 2 Report
Comments and Suggestions for Authors
This study addressed a systematic review of the role of autophagy in adenomyosis. The authors suggested that there is a controversial role in adenomyosis based on the various references published on the topic. Here are some considerations:
- Additional diagrams summarizing the contradictory roles of autophagy may provide greater clarity.
- For each reference, it is necessary to describe more clearly why some studies report activation while others report inhibition.
- From a clinical perspective, this manuscript may better address potential therapeutic implications, such as targeting the autophagy pathway with specific drugs or interventions. A brief discussion of how targeting autophagy could be translated into clinical diagnosis or treatment for adenomyosis-related infertility would add practical value.
- The manuscript would be strengthened by providing specific, actionable recommendations for future research.
Author Response
This study addressed a systematic review of the role of autophagy in adenomyosis. The authors suggested that there is a controversial role in adenomyosis based on the various references published on the topic. Here are some considerations:
- Additional diagrams summarizing the contradictory roles of autophagy may provide greater clarity.
We appreciate the reviewer's suggestion to include additional diagrams summarizing the contradictory roles of autophagy to enhance the clarity of our manuscript. Our manuscript currently contains several diagrams (specifically, Figures 2, 3, and 4) that illustrate the key aspects of autophagy in adenomyosis:
- Figure 2 depicts the regulation of the autophagy pathway, highlighting the molecular mechanisms involved.
- Figure 3 illustrates the cyclic variation of autophagy, estradiol, and progesterone levels across the menstrual cycle, emphasizing how hormonal fluctuations influence autophagy in the endometrium.
- Figure 4 compares autophagy and hormonal regulation in a healthy cycling endometrium versus in adenomyosis, showcasing the dual roles of autophagy in these contexts.
We believe these figures collectively capture the complexities and contradictions associated with autophagy's role in adenomyosis. Adding more diagrams may lead to redundancy and could potentially overwhelm the reader with excessive visual information. Moreover, in the the manuscript's discussion sections, we have elaborated on the contradictory findings, providing clearer explanations for why some studies report activation while others report inhibition of autophagy in adenomyosis. This includes discussing factors such as hormonal influences, the phase of the menstrual cycle, and methodological differences among studies.
- For each reference, it is necessary to describe more clearly why some studies report activation while others report inhibition.
We appreciate the reviewer's insightful comment regarding the need for clearer explanations of why some studies report activation while others report inhibition of autophagy in adenomyosis. We acknowledge that the heterogeneity in these findings can be confusing.
In our systematic review, the articles are organized in Table 1 according to the reported variation of autophagy—either activation or inhibition. Each reference is numbered in the last column of the table and is described in detail in the subsequent paragraphs of the Results section.
We recognize that the reasons for the conflicting results are not entirely understood. However, we have hypothesized that the variation in autophagy reported across studies could be linked to several factors. The first one is the menstrual phase as discussed in lines 272 to 285 of our manuscript. Autophagy levels in the endometrium fluctuate throughout the menstrual cycle due to hormonal changes. Studies collecting samples at different phases may observe varying autophagy activity, contributing to discrepancies in reported results.
Moreover, autophagy is known to be a "double-edged sword," acting either as a protective mechanism or a contributing factor to pathology, depending on the biological context (lines 308 to 312). This dual nature may lead to different interpretations of whether autophagy is upregulated or downregulated in adenomyosis. Variations in experimental designs, sample types (eutopic vs. ectopic endometrium), and detection methods for autophagy markers could also account for the inconsistent findings among studies.
In our conclusion, we emphasize that "the fluctuation of autophagic imbalance throughout the menstrual cycle suggests that the main factor of pathogenesis may rather be its dysfunction regardless of its trend." This statement highlights our belief that it is the disruption of normal autophagic processes, rather than the direction of change, that is significant in adenomyosis pathogenesis.
- From a clinical perspective, this manuscript may better address potential therapeutic implications, such as targeting the autophagy pathway with specific drugs or interventions. A brief discussion of how targeting autophagy could be translated into clinical diagnosis or treatment for adenomyosis-related infertility would add practical value.
We thank the reviewer for highlighting the importance of discussing the clinical implications of our findings. In response, we have added a new subsection to the Discussion (Section 4.6) titled "Therapeutic Implications of Targeting Autophagy in Adenomyosis."
Added to the Discussion:
4.6. Potential Therapeutic Implications of Targeting Autophagy in Adenomyosis
Understanding the multifaceted role of autophagy in adenomyosis opens avenues for potential therapeutic interventions, particularly in addressing fertility issues associated with the condition. Modulating autophagy pathways could offer novel strategies for both the treatment and diagnosis of adenomyosis-related infertility. Autophagy plays a critical role in maintaining cellular homeostasis and has been recognized as a key player in the pathophysiological process related to the endometrium-related diseases such as endometriosis, endometrial carcinoma and infertility (Yang et al. 2019; Shen et al. 2021). Therapeutically, targeting the autophagy pathway may offer a promising avenue for intervention. Pharmacological agents such as rapamycin, an mTOR inhibitor that enhances autophagy, or chloroquine/hydroxychloroquine, which inhibit autophagy, have been explored as potential treatments in similar gynecological diseases (Su et al. 2020) (Nakamura et al. 2024) (Ruiz et al. 2016).
Indeed, preclinical studies have shown that activation of autophagy by rapamycin can reduce the size of endometriotic lesions, suggesting a similar therapeutic potential in adenomyosis (Ren et al. 2016).
Moreover, diagnostic strategies could leverage biomarkers of autophagic activity, such as LC3, Beclin-1 and p62, to better stratify patients and personalize therapeutic approaches (Kong and Yao 2022). Assessing these biomarkers in endometrial tissues could provide insights into the autophagic status and guide clinical decisions.
These perspectives open new avenues for integrating autophagy-targeting therapies into the clinical management of adenomyosis-related infertility. However, further preclinical and clinical studies are necessary to validate these approaches and determine their safety and efficacy.
- The manuscript would be strengthened by providing specific, actionable recommendations for future research.
We appreciate the reviewer's suggestion to include specific, actionable recommendations for future research. In line with this, we have added a new subsection (Section 4.7) titled "Future Directions" to our Discussion.
Added to the Discussion:
4.7. Future directions
The contradictory role of autophagy in adenomyosis underscores the need for further research to clarify its mechanisms and therapeutic potential. Future studies should monitor autophagy levels throughout the menstrual cycle in both eutopic and ectopic endometrial tissues to determine whether autophagic imbalance is consistent or varies with hormonal fluctuations. Elucidating the molecular mechanisms underlying autophagy dysregulation using advanced molecular techniques -such as transcriptomics, proteomics, and metabolomics—is crucial. Understanding the specific signaling pathways and genes involved may reveal novel therapeutic targets and enhance our comprehension of the disease's etiology.
Exploring the therapeutic potential of autophagy modulators remains crucial. Preclinical studies assessing agents like mTOR inhibitors (e.g., rapamycin) and autophagy inhibitors (e.g., chloroquine) should evaluate their efficacy in reducing lesion size, inhibiting cell proliferation, and improving fertility outcomes. Combining these agents with existing hormonal therapies might enhance treatment effectiveness and overcome progesterone resistance, potentially translating into tangible benefits for patients.
Developing autophagy-related diagnostic biomarkers, such as LC3-II, Beclin-1, and p62, offers a promising avenue for early detection and personalized treatment. Validating these biomarkers in clinical settings could facilitate disease monitoring and patient stratification, ultimately improving management strategies for adenomyosis. Additionally, understanding the interaction between autophagy and oxidative stress may provide insights into lesion progression and identify new therapeutic targets. Combining antioxidants with autophagy modulators might offer a synergistic approach to treatment, potentially alleviating symptoms and improving fertility.
Standardizing research methodologies—including unified definitions of adenomyosis types (e.g., focal vs. diffuse) and consistent techniques for measuring autophagy—is vital for advancing the field and improving the reliability of results. Ensuring that future studies specify the menstrual cycle phase during sample collection is essential, given the significant variation of autophagy throughout the cycle. By addressing these areas, future research can deepen our understanding of autophagy's role in adenomyosis and pave the way for developing targeted therapies to improve fertility outcomes for affected women.
Reference List
Kong, Z., and T. Yao. 2022. 'Role for autophagy-related markers Beclin-1 and LC3 in endometriosis', BMC Womens Health, 22: 264.
Nakamura, A., Y. Tanaka, T. Amano, A. Takebayashi, A. Takahashi, T. Hanada, S. Tsuji, and T. Murakami. 2024. 'mTOR inhibitors as potential therapeutics for endometriosis: A narrative review', Mol Hum Reprod.
Ren, X. U., Y. Wang, G. Xu, and L. Dai. 2016. 'Effect of rapamycin on endometriosis in mice', Exp Ther Med, 12: 101-06.
Ruiz, A., S. Rockfield, N. Taran, E. Haller, R. W. Engelman, I. Flores, P. Panina-Bordignon, and M. Nanjundan. 2016. 'Effect of hydroxychloroquine and characterization of autophagy in a mouse model of endometriosis', Cell Death Dis, 7: e2059.
Shen, Hui-Hui, Tao Zhang, Hui-Li Yang, Zhen-Zhen Lai, Wen-Jie Zhou, Jie Mei, Jia-Wei Shi, Rui Zhu, Feng-Yuan Xu, Da-Jin Li, Jiang-Feng Ye, and Ming-Qing Li. 2021. 'Ovarian hormones-autophagy-immunity axis in menstruation and endometriosis', Theranostics, 11: 3512-26.
Su, Y., J. J. Zhang, J. L. He, X. Q. Liu, X. M. Chen, Y. B. Ding, C. Tong, C. Peng, Y. Q. Geng, Y. X. Wang, and R. F. Gao. 2020. 'Endometrial autophagy is essential for embryo implantation during early pregnancy', J Mol Med (Berl), 98: 555-67.
Yang, S., H. Wang, D. Li, and M. Li. 2019. 'Role of Endometrial Autophagy in Physiological and Pathophysiological Processes', J Cancer, 10: 3459-71.
Round 2
Reviewer 2 Report
Comments and Suggestions for Authors
I think the authors have adequately responded to the comments presented.